# Cost-benefit analysis of a trifocal intraocular lens versus a monofocal intraocular lens from the patient's perspective in the United States

**John Berdahl**[1][☉]*, **Chandra Bala**[2][☉], **Mukesh Dhariwal**[3][‡], **Hemant Rathi**[4][‡], **Ritu Gupta**[5][‡]

**1** Vance Thompson Vision, Sioux Falls, South Dakota, United States of America, **2** personalEYES, Sydney, New South Wales, Australia, **3** Global Health Economics & Outcomes Research, Alcon Vision LLC, Fort Worth, Texas, United States of America, **4** Skyward Analytics Pte. Ltd., Singapore, Singapore, **5** Skyward Analytics Pvt. Ltd., Gurgaon, Haryana, India

☉ These authors contributed equally to this work.
‡ These authors also contributed equally to this work.
* johnberdahl@gmail.com

## Abstract

### Purpose

To conduct a cost-benefit analysis of AcrySof IQ PanOptix trifocal intraocular lens (TFNT00 IOL) versus AcrySof monofocal IOL (SN60AT) from the patient perspective in the United States (US).

### Methods

A de novo Markov model was developed to estimate the mean total lifetime patient costs and vision-related quality of life (measured as quality adjusted life-years (QALYs)) with each intervention (TFNT00 and SN60AT) and the incremental differences between these two treatments. The resulting incremental quality of life gain was mapped to the US patient willingness to pay threshold of $50,000 per QALY gain to estimate the lifetime net monetary value, measured as the net monetary benefit of TFNT00 IOL. Model inputs (transition probabilities, costs, discount rate, utilities, and event rates) were derived from the FDA IDE study (NCT03280108), published literature, clinical experience, and other relevant sources.

### Results

Bilateral cataract surgery with implantation of the advanced technology IOL (AT-IOL) TFNT00 provides improved vision-related quality of life (QALY gain of 0.67) at an incremental lifetime cost of $2,783 compared to monofocal IOL. This incremental QALY gain translated into a lifetime net monetary benefit of $30,941 at the patient willingness to pay threshold of $50,000/QALY gain. Results were most sensitive to disutility due to wearing glasses, patient out of pocket costs for bilateral AT-IOL procedure, and post-operative spectacle dependence rates.

**Data Availability Statement:** All relevant data are within the paper and its Supporting Information files.

**Funding:** study was funded by Alcon Vision LLC, Fort Worth, Texas, USA, the manufacturer of AcrySof IQ PanOptix and AcrySof SN60AT IOLs. JB and CB received consulting fees from Alcon. MD is a full-time employee of Alcon Vision LLC (the study sponsor). HR and RG are employees of Skyward Analytics Pte. Ltd. and Skyward Analytics Pvt. Ltd., respectively, and received consulting fees from Alcon to conduct this study. All authors supported study design, data collection, analysis, and preparation of the manuscript. No authors received remuneration for writing this manuscript. The authors report no other conflicts of interest related to this work.

**Competing interests:** This study was sponsored by Alcon Vision LLC, Fort Worth, Texas, USA (https://www.alcon.com/). JB and CB received consulting fees from Alcon. MD is an employee of Alcon Vision LLC (the study sponsor). HR and RG are employees of Skyward Analytics Pte. Ltd. and Skyward Analytics Pvt. Ltd., respectively, and received consulting fee from Alcon to conduct this study. The authors report no other conflicts of interest related to this work.

## Conclusions

AcrySof IQ PanOptix IOL provides greater improvement in vision related quality of life compared to no presbyopia correction with a monofocal IOL. This study shows PanOptix is a cost-beneficial treatment strategy for patients willing to pay out of pocket for cataract surgery.

## Introduction

Globally, cataract surgery is one of the most commonly performed surgical procedures, and could well be considered among the most successful treatments in the field of medicine [1,2]. It is also one of the most effective health care interventions that provides greater health outcomes for a majority of patients at relatively lower costs [3–5].

The United States (US) Centers for Medicare and Medicaid Services (CMS) covers conventional cataract surgery with a monofocal intraocular lens (IOL) [6]; however, these IOLs are generally only effective in correcting distance vision and patients may require spectacles for intermediate and/or near vision following surgery [7]. Alternatively, advanced technology IOLs (AT-IOLs) such as presbyopia-correcting IOLs were designed in recent years to offer patients complete spectacle independence among all distances (far, intermediate, and near vision) [7], but CMS has ruled that patients must pay additional out of pocket costs for this advanced technology [8]. In order for patients to make informed decisions for what IOL to choose, it is imperative patients understand the clinical and monetary benefits premium AT-IOLs can offer and the potential positive impact these new technologies can provide in relation to improving patient vision-related quality of life.

In 2019, the US Food and Drug Administration (FDA) approved the first-ever trifocal IOL, AcrySof IQ PanOptix (model TFNT00; Alcon) [9,10]. The pivotal PanOptix IOL FDA IDE clinical trial (NCT03280108) conducted across the US demonstrated continuous range of vision from distance to near, with increased spectacle independence and a high level of patient satisfaction, as compared to the AcrySof IQ monofocal IOL (model SN60AT; Alcon) [9,10]. Some studies have reported economic evaluations of AT-IOL versus monofocal IOL options [3,11,12]; however, there currently is no published evidence on the cost benefits of TFNT00 IOL in the US. Consequently, the objective of this study was to conduct a cost-benefit analysis of bilateral cataract surgery with TFNT00 trifocal IOL versus SN60AT monofocal IOL from the US patient perspective.

## Methods

### Model structure

A de novo Markov model (Fig 1) was developed to simulate a patient's remaining lifetime after cataract surgery, and to measure the cost-benefit of IOL treatments when implanted with TFNT00 versus SN60AT. The median patient age at baseline used in this model (68 years old) was sourced from the FDA IDE clinical study [10]. Patient health states in the model were based upon plausible health states post cataract surgery, which included: well (patients with complete spectacle independence and absence of visual disturbances), spectacle dependent (patients requiring reading, distance, bifocal, or progressive glasses), visual disturbances (patients with moderate to very bothersome glare/haloes/starbursts)–with or without spectacles, and death. Death was an absorbing health state in the model. Rates for lens explantation and YAG capsulotomy procedures were also included as post-surgery events in the model. The

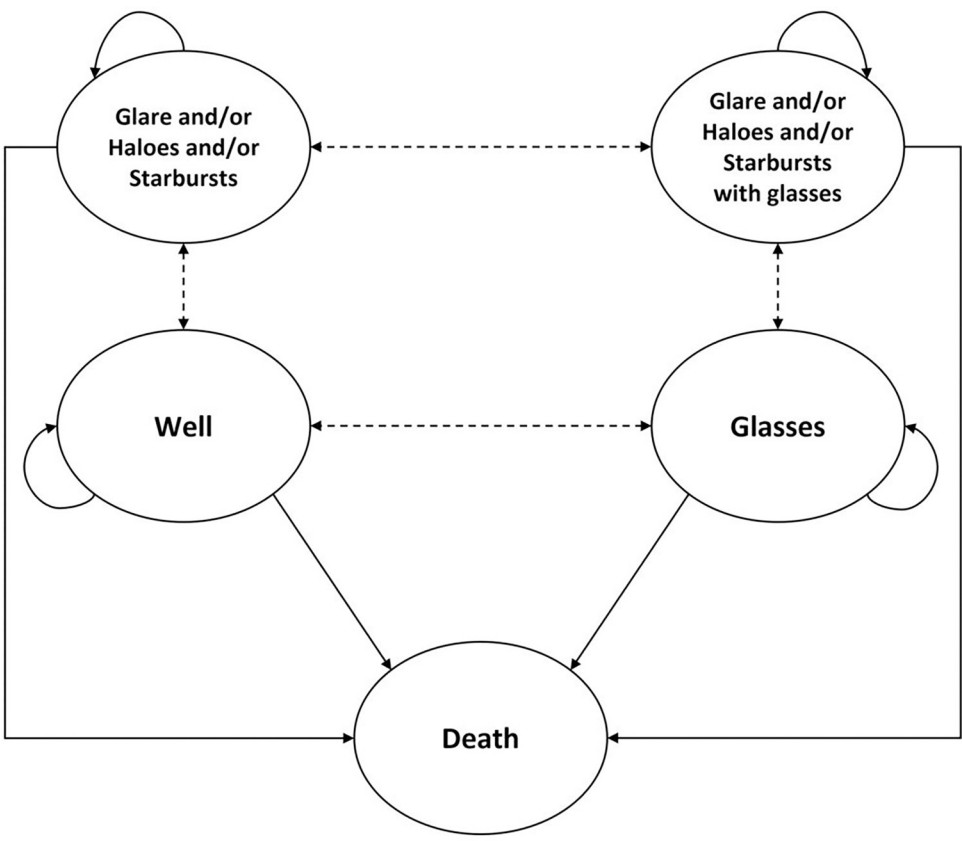

**Fig 1. Markov model structure.**

model was developed from the patient's perspective, therefore all cost inputs reflected patient out of pocket costs.

## Model inputs

**Clinical inputs.** Transition probabilities were used to map out the different health states patients may experience post-cataract surgery, and are applied one time in the first yearly cycle of the model, except for resolution of visual disturbances which is applied in the second cycle, following the first year (see Table 1). Overall spectacle dependence rates were 19.5% and 92% for TFNT00 and SN60AT IOLs, respectively [10]. Distribution of spectacle dependence type

**Table 1. Transition probabilities.**

| Parameter | Application in model | TFNT00 IOL | SN60AT IOL | References |
|---|---|---|---|---|
| Overall spectacle dependence | Within 1 year | 0.195 | 0.920 | Modi et al [10] |
| Moderate to very bothersome glare and/or haloes and/or starbursts | Within 1 year | 0.120 | 0.072 | AcrySof IQ PanOptix Directions For Use [15] |
| Resolution of glare/haloes/starbursts | After 1 year | 0.810 | 0.810 | Hu et al [11] |
| IOL explantation | Within 1 year | 0.008 | 0.009 | Modi et al [10] |
| YAG capsulotomy | Within 1 year | 0.248 | 0.061 | Data on file [13] |

IOL = intraocular lens; YAG = yttrium aluminum garnet.

**Table 2. Distribution of patients by type of spectacle need who were spectacle dependent post-surgery (pre-LASIK)\*.**

| Parameter | TFNT00 IOL | SN60AT IOL | References |
|---|---|---|---|
| Reading glasses | 75.0% | 40.0% | Data on file [13] and clinical input |
| Distance glasses | 20.0% | 10.0% | Data on file [13] and clinical input |
| Bifocal glasses | 2.5% | 25.0% | Data on file [13] and clinical input |
| Progressive glasses | 2.5% | 25.0% | Data on file [13] and clinical input |

\*Distributions were normalized to equal 100%.

IOL = intraocular lens; LASIK: Laser-assisted in situ keratomileusis.

by vision correction needs (near, intermediate, and/or distance) after TFNT00 and SN60AT IOL implantation was derived from internal data and clinical experience, and normalized to equal 100% (see Table 2) [13]. Furthermore, patients opting for AT-IOL cataract surgery in the US may receive a LASIK surgery enhancement to maximize the probability of achieving 20/20 uncorrected near, intermediate, and distance visual acuity outcomes and it is often bundled into the AT-IOL procedure package. Therefore, of the 14.6% TFNT00 patients who did not achieve near vision spectacle independence post-surgery, an estimated 50% were assumed to receive LASIK enhancement, based on clinical experience. Similarly, of the TFNT00 patients who did not achieve distance (3.9%) and/or intermediate (1%) spectacle independence, approximately 75% were assumed to undergo LASIK enhancement; LASIK effectiveness rate was set at 90% [14]. In accordance with observed clinical experience, spectacle dependence/independence achieved post-surgery and post-LASIK procedure remained constant for the patient's remaining lifetime.

The proportion of patients experiencing moderate to very bothersome visual disturbances (glares/halos/starbursts) were 12.0% and 7.2% for TFNT00 and SN60AT, respectively [15]. The resolution rate of glare/haloes/starbursts (81%) was obtained from Hu et al study [11]. The probability of visual disturbances developing and resolving were assumed to be similar for both spectacle-independent and spectacle-dependent patients. Similar to the Hu et al study, it was expected that chances of developing bothersome visual disturbances would occur within the first yearly cycle post-surgery [11].

The probability of death was estimated based upon mortality data from the Centers for Disease Control and Prevention (CDC) National Center of Health Statistics Life Tables 2017 [16], and were applied to the proportion of alive patients in each health state of each cycle. The mortality rate was assumed to be the same across all the health states for both TFNT00 and SN60AT IOL arms in the model.

**Cost inputs.** The costs considered in the model included bilateral implantation of TFTN00 IOL and SN60AT IOL procedures, post-surgery ocular medications, IOL explantation, YAG laser capsulotomy, glasses, and optometrist visit. For Medicare covered benefits (conventional monofocal cataract surgery with SN60AT IOL, eye drops, IOL explantation, YAG laser capsulotomy, and optometrist visits), the average patient coinsurance was estimated to be 15% in the model; this takes into consideration that most patients pay 20% for Medicare-covered services and 35% of those Medicare patients receive supplemental insurance (Medigap) that helps cover additional costs [17,18]. AT-IOLs are not covered under Medicare and costs borne by the patient can range from $1,500 to $4,000 per eye, depending on the type of IOL and where in the US the procedure is done [19]. Therefore, based on this literature and clinical experience, the cost of bilateral TFNT00 AT-IOL procedure was estimated to be $6,000 in our base case analysis, with variations in out of pocket costs addressed in the sensitivity analysis.

For those requiring LASIK surgery, the cost of this procedure was included as part of the AT-IOL procedure package cost. The costs of glasses were obtained from publicly available sources and internal data [20,21]. It's recommended that adults 65 years and older should get an eye exam every year, which may lead to an updated prescription and new glasses [22]; additionally, patients may desire new glasses if they want different frames, a newer style, or their previous ones broke [22]. Based on this knowledge and clinical experience, patients were assumed to receive a new pair of glasses every 15 months. Full details of cost inputs are provided in Table 3.

**Dis-utilities.**   In this model, the "well" health state represents patients with optimal vision-related health (complete spectacle independence at all distances as proxy for optimized near, intermediate, and far distance visual acuity and absence of any visual disturbances). Therefore, the vision-related utility of the "well" health state was assumed to be 1, which reflects a similar approach to the Hu et al study [11].

Disutility associated with adverse events of bothersome visual disturbances (glare/haloes/starbursts, -0.18), wearing glasses (-0.065), and undergoing lens explantation procedure (-0.150) were previously reported in Brown et al [25], Dobrez and Calhoun [26], and Busbee et al [4], respectively and were used in this analysis. Patients were assumed to experience bothersome visual disturbances (glare/haloes/starbursts) for an estimated 4 hours per day (60.9 days in a calendar year), based on increased reports of visual disturbances under dim light conditions, particularly in the night time [27].

**Table 3. Cost inputs.**

| Parameter | Cost ($) | References/Notes |
|---|---|---|
| **Cost of bilateral SN60AT IOL procedure** | $516.89 | CPT codes 66984, 92004; Estimating 15% patient coinsurance per Medicare/Medigap [17,18] |
| **Cost of bilateral TFNT00 IOL procedure** | $6,000.00 | All About Vision [19] and clinician input |
| **Total cost of IOL explantation** | $309.93 | CPT codes 66986, 92014; Estimating 15% patient coinsurance per Medicare/Medigap [17,18] |
| **Cost of eye drops** | | |
| Ketorolac ophthalmic (5mL) | $65.66 | Drugs.com [23] |
| Prednisolone acetate (5mL) | $31.55 | Drugs.com [23] |
| Moxifloxacin hydrochloride (3mL) | $57.59 | Drugs.com [23] |
| Total cost | $154.80 | |
| **Total cost of eye drops** | $23.22 | Estimating 15% patient coinsurance per Medicare/Medigap [17,18] |
| **Total cost of YAG laser capsulotomy** | $66.91 | CPT codes 66821, 92014; Estimating 15% patient coinsurance per Medicare/Medigap [17,18] |
| **Cost of glasses** | | |
| Reading glasses | $14.95 | Walmart.com (TiFlex Prescription Glasses, T1039 Dark Pewter) [21] |
| Distance glasses | $126.00 | Walmart.com (VWE Rectangular Metal Frame Reading Glasses with Spring Hinge) [21] |
| Bifocal glasses | $ 500.00 | Market research data [20] |
| Progressive glasses | $ 500.00 | Market research data [20] |
| **Cost of optometrist visit** | **$11.42** | CPT code 99213 –Medicare national unadjusted payment rate for 2020 [24]; Estimating 15% patient coinsurance per Medicare/Medigap [18,19] |

CPT = current procedural terminology; IOL = intraocular lens; YAG = yttrium aluminum garnet.

## Base-case analysis

A lifetime horizon of 30 years with a yearly cycle length was used in the base-case analysis. Using the CDC life tables [16], the 30-year time horizon simulates the patient's remaining lifetime post-cataract surgery to fully capture the improvement in patient quality of life. A half-cycle correction was applied to estimate the average time spent in each health state.

In the base-case analysis, total lifetime costs and quality adjusted life years (QALYs) were estimated based on the median patient age of 68 in the FDA IDE clinical study [10]. The model also estimated the change in costs and change in QALYs among the two interventions (TFNT00 and SN60AT). In addition, the net monetary benefit (NMB) that patients could expect from TFNT00 was calculated as the incremental QALY benefit multiplied by the patient willingness-to-pay (WTP) per QALY gain minus incremental costs. Future costs and outcomes were discounted at an annual rate of 3%.

## Sensitivity analysis

To understand the effect of uncertainty around the model input parameters on the base-case results and to identify key drivers affecting the results, three different types of sensitivity analyses were conducted: one-way sensitivity analysis (OWSA), probabilistic sensitivity analysis (PSA), and scenario analysis. OWSA was performed to assess the impact of change in individual parameters on the model results. All parameters were varied within reported or published 95% confidence intervals (CIs). For input parameters without reported or published 95% CI, the mean, standard deviation, standard error, and/or the number of observations were used in estimating 95% CI.

PSA with 1,000 simulations was conducted to examine the effect of uncertainty around all model inputs simultaneously. In each simulation, a value was drawn for every input parameter by random sampling based on their respective distributions. For model parameters bounded by 0 and 1, the beta distribution was used to vary the parameter using the reported mean and the number of observations. For cost and other parameters whose values are greater than 0, the log-normal distribution was used to vary the parameter based on the mean and standard error of the parameter. If standard error was not available, it was assumed to equal 20% of the mean value of the parameter. Normal distribution was used to vary disutilities. Dirichlet distribution was used for multivariate probability parameters.

**Scenario analysis.** Several scenario analyses were performed to understand the effect of various model settings or assumptions on model results. The following model settings were varied in the model: time horizon (5, 10, 15, 20, and 25 years), patients' age (65 years old), and discount rates (0% and 5%).

# Results

## Base-case results

In the base-case analysis with a lifetime (30-year) horizon, total quality adjusted life-years (QALYs) per patient were 13.09 and 12.41 for TFNT00 and SN60AT IOLs, respectively. The total lifetime costs incurred by patients implanted with TFNT00 and SN60AT IOLs were $6,137 and $3,354, respectively. Bilateral implantation of TFNT00 IOL provided greater vision-related quality of life (0.67 QALY gained) at an incremental lifetime cost of $2,783 compared to the SN60AT IOL. The model predicted that patients implanted with bilateral TFNT00 IOL would have an average lifetime spectacles cost savings of $2,593 compared to those implanted with SN60AT IOL. At a willingness-to-pay (WTP) threshold of $50,000 per QALY gain, lifetime net monetary benefit (NMB) per patient with bilateral TFNT00 IOL implantation was $30,941. Disaggregated costs and QALYs associated with different health states with each intervention are presented in Table 4.

**Table 4. Disaggregated base-case results.**

| Parameters | TFNT00 IOL | SN60AT IOL | Incremental Outcome |
|---|---|---|---|
| **Costs** | | | |
| **IOL procedure cost** | **$6,000** | **$517** | **$5,483** |
| **IOL explantation cost** | **$2** | **$3** | **$0** |
| **Eye drops cost** | **$23** | **$23** | **$0** |
| **YAG surgery cost** | **$17** | **$4** | **$12** |
| **Optometrist's visit cost** | **$14** | **$133** | **-$119** |
| **Glasses cost: Total** | **$81** | **$2,674** | **-$2,593** |
| Reading glasses | $49 | $223 | -$174 |
| Distance glasses | $16 | $118 | -$101 |
| Bifocal glasses | $8 | $1,167 | -$1,159 |
| Progressive glasses | $8 | $1,167 | -$1,159 |
| **Total cost** | **$6,137** | **$3,354** | **$2,783** |
| **Life-Years** | | | |
| Well | 11.61 | 1.50 | 10.12 |
| Glasses only | 1.18 | 11.45 | -10.27 |
| Glare/haloes | 0.35 | 0.02 | 0.33 |
| Glare/haloes with glasses | 0.04 | 0.21 | -0.17 |
| **Total Life-Years** | **13.18** | **13.18** | **0.00** |
| **QALYs** | | | |
| Well | 11.61 | 1.50 | 10.12 |
| Glasses only | 1.11 | 10.71 | -9.60 |
| Glare/haloes | 0.34 | 0.02 | 0.32 |
| Glare/haloes with glasses | 0.03 | 0.19 | -0.16 |
| IOL explantation | -0.001 | -0.001 | 0.00 |
| **Total QALYs** | **13.09** | **12.41** | **0.67** |

IOL = intraocular lens; QALY = quality-adjusted life-years; YAG = yttrium aluminum garnet.

### Sensitivity analysis

The one-way sensitivity analysis (OWSA) identified that NMB results were most sensitive to the disutility due to wearing glasses, cost of bilateral AT-IOL procedure, and post-operative spectacle dependence rates (see Fig 2).

Results from the probabilistic sensitivity analysis (PSA) confirmed the robustness of the base-case deterministic results as the average NMB from 1,000 simulations was estimated to be $30,875, which was similar to the deterministic base-case NMB. The cost-effectiveness acceptability curve presented in Fig 3 projected that at a WTP per QALY gain threshold of $13,000 and above, the probability of TFNT00 IOL yielding positive NMB was 100%.

**Scenario analyses.** The results of the scenario analyses are presented in Fig 4. The NMB was most affected by changing the time horizon to 5 and 10 years; however, the scenario analysis results showed that TFNT00 IOL would still yield a NMB for patients at these shorter time horizons. In addition, scenario analyses showed higher NMB for younger age patients.

### Discussion

This study evaluated the cost-benefits of the PanOptix trifocal IOL from the patient's perspective in the US. To the authors' knowledge, this the first attempt to estimate the cost-benefits of the PanOptix trifocal IOL technology. Our model findings show bilateral AT-IOL cataract

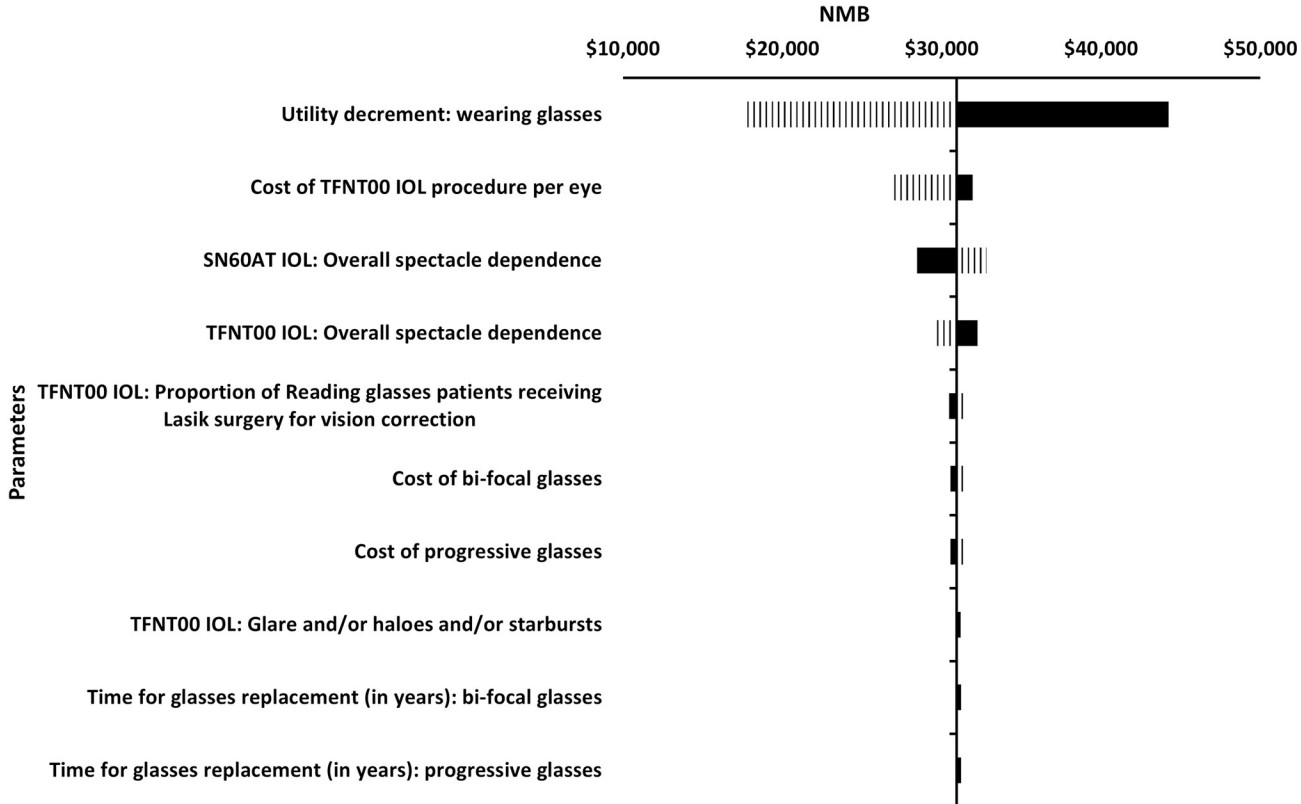

**Fig 2. Tornado diagram summarizing the OWSA results.** ■ Lower Bound ‖ Upper Bound. Abbreviations: IOL = intraocular lens; NMB = net monetary benefit; OWSA = one-way sensitivity analysis.

surgery with PanOptix IOL leads to improvement in patient's vision-related quality of life versus no presbyopia correction with a conventional monofocal IOL. These improved quality of life benefits translated into a lifetime net monetary benefit (NMB) of nearly $31,000 for patients, using the commonly used US patient WTP threshold [28]. This represents a 5-fold return on the patient's out of pocket investment for bilateral implantation of PanOptix.

Our study has several strengths. First, we followed the recommended methodology and the reference case proposed by the US Institute for Clinical and Economic Review (ICER) to develop this analysis [29]. ICER suggests a range of $50,000-$150,000/QALY gain for the WTP threshold for the economic evaluation of novel treatments [28]. The WTP threshold of $50,000/QALY was used in this study to reflect the patient perspective of the health care system, bearing in mind that patients nowadays are willing to pay up to $100,000-$200,000/QALY [30]. Using the higher WTP threshold of $150,000, NMB associated with bilateral AT-IOL cataract surgery with PanOptix IOL would further increase to $98,388. Therefore, the return on investment could be 16-fold greater than the patient out of pocket costs for the procedure.

Moreover, model structure and key assumptions such as the resolution rate of visual disturbances used in the present study were broadly similar to the recent multifocal cost-effectiveness analysis conducted in the US [11]. In addition, the pivotal efficacy estimates (spectacle dependence and bothersome visual disturbances rates) were obtained from the FDA IDE clinical study in which outcomes were directly measured in the clinical study using validated patient reported questionnaires [9,10,31].

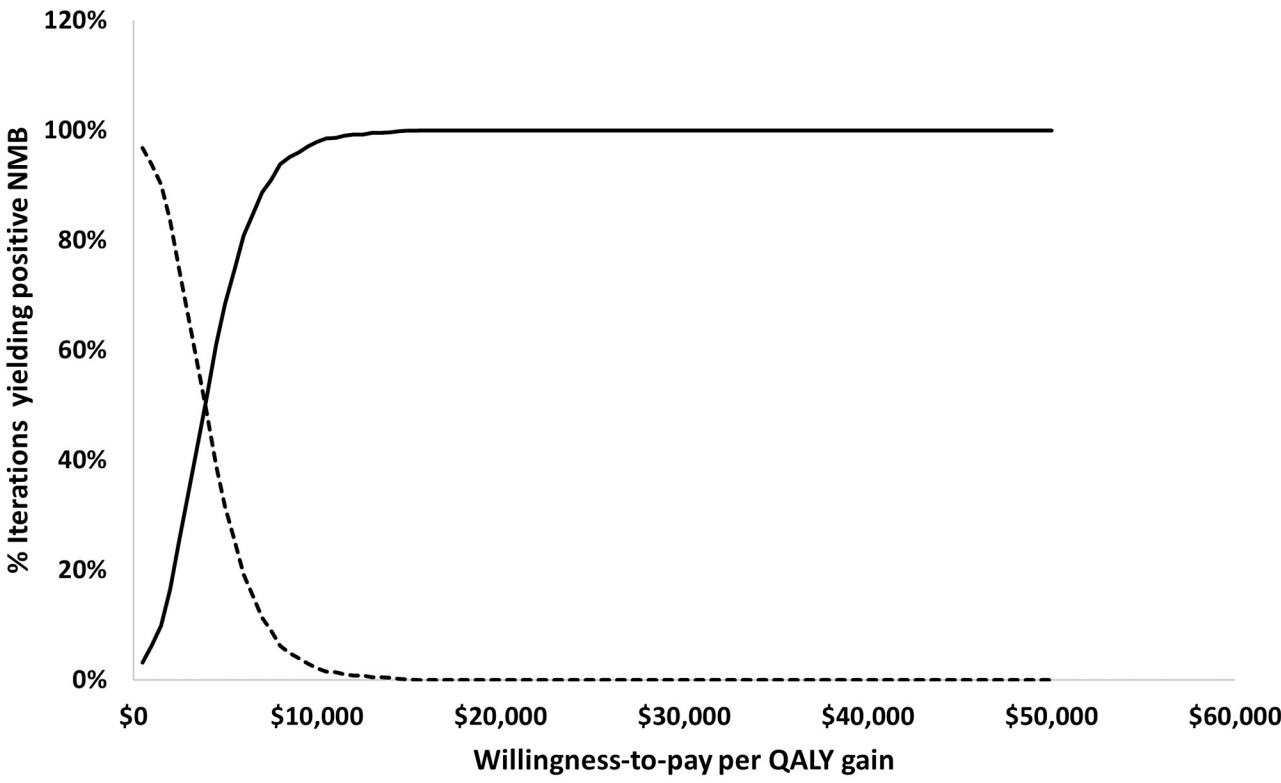

**Fig 3. Cost-effectiveness acceptability curve.** Solid line: TFNT00 IOL, Dashed line: SN60AT IOL, Abbreviations: IOL = intraocular lens; NMB = net monetary benefit; QALY = quality adjusted life years.

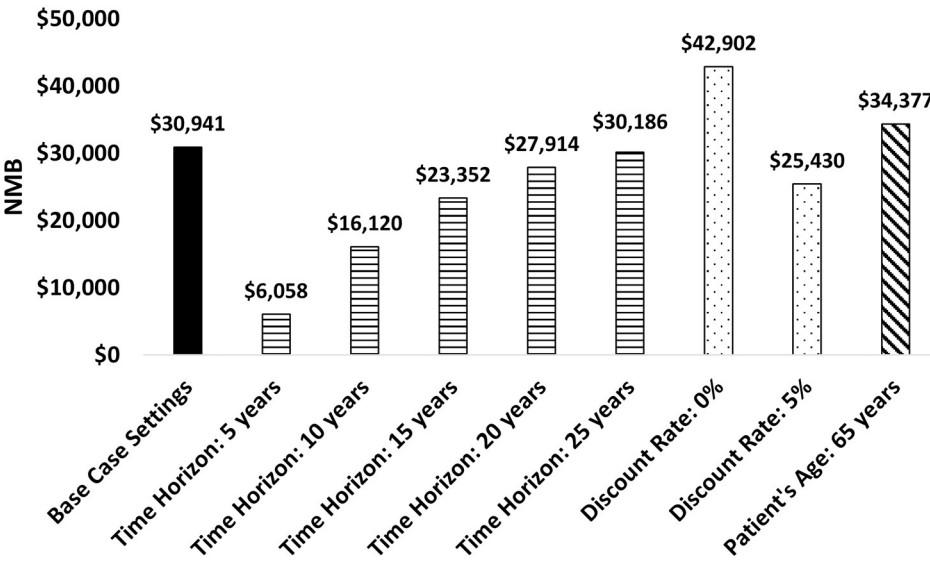

**Fig 4. Scenario analysis results.** Abbreviations: NMB = net monetary benefit.

One of the key findings of this study include the model's sensitivity to the patient out of pocket procedure costs. In the base case analysis, patient out of pocket costs for bilateral implantation of PanOptix was set at $6,000. However, procedure costs are independently set by each facility and therefore could vary substantially. Thus, we tested this in the one-way sensitivity analysis (OWSA) (Fig 2), highlighting that even doubling the patient out of pocket costs to $12,000, the NMB per patient for bilateral implantation of PanOptix would still be $24,941, which is an approximate 2-fold greater return on the patient's investment.

There are some limitations to our analysis. First, we conducted this analysis for patients' remaining lifetime (a maximum of 30 years), however, we did not have the long-term clinical efficacy data to inform clinical input parameters in the model. Future studies should be considered to evaluate the long-term efficacy of multifocal IOLs. In addition, we relied on independent publications to source dis-utilities for different health states which is an acceptable alternative in the absence of direct estimates from the clinical study population. Average patient coinsurance for procedures was assumed to be 15% of the Medicare approved payment in our analysis. An independent report suggests that in 2017, approximately 35% CMS beneficiaries had Medicare supplemental (Medigap) coverage to help cover patient coinsurance costs [17]. However, we believe a 15% coinsurance assumption would still be a fair representation of the national average. Cost of glasses was also estimated from publicly available sources and internal market research data which may be conservative estimates since the spectacle costs can vary significantly for each patient depending upon the type of selected frame, optical prescription pattern, and other features such as blue light filtering. Nevertheless, we believe conservative estimates did not bias the results in favor of the intervention (TFNT00).

Moreover, surgeon real-world clinical experience was used to fill gaps in input parameters to estimate the rate and effectiveness of LASIK enhancement as well as the development and resolution rates of visual disturbances. Authors recommend generating clinical evidence on these parameters in future studies. However, such limitations were not unique for this analysis, rather they are common in health economic evaluation for innovative, newly launched medical technologies. In fact, we conducted various sensitivity analyses to estimate the impact of uncertainty in the model parameters, and the model results were robust with respect to many settings/assumptions tested under plausible scenario analyses. The robustness of the results was further confirmed by the probability sensitivity analysis; the average results from 1,000 simulations was similar to the deterministic base case results.

It is important to note that our analysis did not estimate the incremental quality of life and cost benefits of conventional cataract surgery compared to no surgery. However, a previous study estimated an incremental lifetime QALY gain of 2.82 with bilateral cataract surgery compared to no surgery [32]. This is indirectly substantiated in our study as the analysis shows that patients who received the conventional cataract surgery with a monofocal IOL (SN60AT) did accumulate substantial QALYs over lifetime. These points make a strong case for the continuous coverage for conventional cataract surgery at prevalent CMS payment rates for facilities and surgeons. Our analysis also supports the established patient self-pay pathway for patients who opt for the refractive benefits of AT-IOLs and desire greater improvement in their quality of life. We recommend that patients undergoing cataract surgery should be provided information on the clinical and cost benefits of PanOptix IOL in order to make informed treatment choices.

## Conclusions

In conclusion, this analysis demonstrates that AcrySof IQ PanOptix trifocal IOL provides greater improvement in vision-related quality of life compared to implantation of standard

monofocal IOL during cataract surgery. This improved quality of life provides lifetime net monetary benefits for patients in the US and is considered a cost-beneficial treatment strategy for patients who wish to pay out of pocket for cataract surgery.

## Supporting information

**S1 File. Detailed calculations of the cost-benefit study.**
(PDF)

## Acknowledgments

The authors would like to thank Amit Gupta and Shantanu Jawla (Skyward Analytics Pvt. Ltd.) for their support in the model development and writing of the manuscript, and Kayla Mills (Alcon Vision LLC, USA) for managing and providing writing assistance towards the development of this manuscript.

## Author Contributions

**Conceptualization:** John Berdahl, Chandra Bala, Mukesh Dhariwal.

**Data curation:** John Berdahl, Chandra Bala, Mukesh Dhariwal, Hemant Rathi, Ritu Gupta.

**Formal analysis:** John Berdahl, Chandra Bala, Mukesh Dhariwal, Hemant Rathi, Ritu Gupta.

**Funding acquisition:** Mukesh Dhariwal.

**Investigation:** Hemant Rathi, Ritu Gupta.

**Methodology:** John Berdahl, Chandra Bala, Mukesh Dhariwal, Hemant Rathi, Ritu Gupta.

**Project administration:** John Berdahl, Chandra Bala, Mukesh Dhariwal.

**Resources:** Mukesh Dhariwal, Hemant Rathi, Ritu Gupta.

**Software:** Hemant Rathi, Ritu Gupta.

**Supervision:** John Berdahl, Chandra Bala, Mukesh Dhariwal.

**Validation:** John Berdahl, Chandra Bala, Mukesh Dhariwal, Hemant Rathi, Ritu Gupta.

**Visualization:** Mukesh Dhariwal, Hemant Rathi, Ritu Gupta.

**Writing – original draft:** Hemant Rathi, Ritu Gupta.

**Writing – review & editing:** John Berdahl, Chandra Bala, Mukesh Dhariwal.

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
