## [Decision Letter · Decision Letter 0]

4 Aug 2022

PONE-D-22-16771Cost-benefit analysis of a trifocal versus a monofocal intraocular lens from the patient's perspective in the United StatesPLOS ONE

Dear Dr. Berdahl,

Thank you for submitting your manuscript to PLOS ONE. After careful consideration, we feel that it has merit but does not fully meet PLOS ONE’s publication criteria as it currently stands. Therefore, we invite you to submit a revised version of the manuscript that addresses the points raised during the review process.

We look forward to receiving your revised manuscript.

Kind regards,

Guangming Jin, M.D.

Academic Editor

PLOS ONE

Journal Requirements:

" This study was sponsored by Alcon Vision LLC, Fort Worth, Texas, USA (https://www.alcon.com/). JB and CB received consulting fees from Alcon. MD is an employee of Alcon Vision LLC (the study sponsor). HR and RG are employees of Skyward Analytics Pte. Ltd. and Skyward Analytics Pvt. Ltd., respectively, and received consulting fee from Alcon to conduct this study. The authors report no other conflicts of interest related to this work."

Additional Editor Comments:

Based on the comments from reviewers, major revision is recommended. Please make a point to point response to each comments.

Reviewers' comments:

Reviewer's Responses to Questions

**Comments to the Author**

1. Is the manuscript technically sound, and do the data support the conclusions?

Reviewer #1: No

Reviewer #2: Partly

2. Has the statistical analysis been performed appropriately and rigorously? 

Reviewer #1: No

Reviewer #2: I Don't Know

3. Have the authors made all data underlying the findings in their manuscript fully available?

Reviewer #1: No

Reviewer #2: Yes

4. Is the manuscript presented in an intelligible fashion and written in standard English?

Reviewer #1: Yes

Reviewer #2: Yes

5. Review Comments to the Author

Reviewer #1: Comments to authors

Thank you for the opportunity to review this manuscript. In this study, the authors conduct a cost-effectiveness analysis to evaluate the cost-effectiveness of AcrySof IQ PanOptix intraocular lens (TFNT00 IOL) relative to AcrySof monofocal IOL (SN60AT) from the US patient perspective. They do this by conducting a Markov model informed by published data: clinical trial, cost, and HRQOL, in addition to assumption/clinical input. In their base case, they argue that TFNT00 IOL is cost-effective to patients relative to SN60AT based on a willingness-to-pay per QALY threshold of $50,000/QALY. In sensitivity analyses, they find their results to be robust to changes in select model parameters.

The manuscript writing is in general well-written and easy to follow. However, a severe limitation of the paper is reliance on assumption/clinical input: the majority of input parameters for the study are assumed. For instance, the entirety of Table 2 is based on “clinical input.” I recommend the authors conduct literature searches and/or primary data analyses to generate reasonable parameter estimates in lieu of powering the model almost entirely based on assumption/clinical opinion.

Specific comments

1. The article often refers to the analysis as a “cost-benefit” analysis, but the way the study is conducted, “cost-effectiveness” analysis is more appropriate.

2. Cost-effectiveness analyses from the patient perspective are uncommon. How is the threshold of $50,000 willingness to pay per quality-adjusted life year (QALY) informed?

3. Have the authors conducted a literature review of previous economic evaluations in this disease area? A paragraph in the introduction section should discuss this.

4. Model inputs – the transition probabilities in Table 1 do not appear to be converted to fit the Markov model (i.e., converted to yearly cycles).

5. Disutilities – a utility value of 1 (equivalent to perfect health) in the well state is not clinically reasonable given the advanced age and comorbidity status of the patient population of interest.

6. Probabilistic sensitivity analyses – how were distributions and parameters chosen for each input? What are these distributions?

7. Scenario analyses – how do the authors reconcile scenario analyses starting at ages 50, 55, and 60 if many inputs are Medicare-specific?

8. Discussion – the Institute for Clinical and Economic Review is an independent organization. It is not appropriate to follow their guidelines for economic evaluation in academic work. I would recommend the authors follow recommendations of the Second Panel in Cost-Effectiveness in Health and Medicine instead.

Reviewer #2: This study attempts to validate the implantation of the Alcon PanOptix IOL based on QALY methodology. The problem in this is the unknown long-term quality of vision with this lens, knowing that the material is subject to the development of subsurface nanoglistenings and, to a lesser extent, glistenings. Furthermore, long-term refractive stability is uncertain, knowing the ongoing change in corneal astigmatism with aging.

There are a large number of editing errors in the manuscript.

6. PLOS authors have the option to publish the peer review history of their article (what does this mean?). If published, this will include your full peer review and any attached files.

Reviewer #1: No

Reviewer #2: No

---

## [Author Response · Author response to Decision Letter 0]

15 Sep 2022

We thank you and the reviewers for considering our manuscript entitled “Cost-benefit analysis of a trifocal intraocular lens versus a monofocal intraocular lens from the patient's perspective in the United States” (ID PONE-D-22-16771) and providing valuable comments to further improve this manuscript.

We have made efforts to address reviewer comments and revised the manuscript accordingly. Please refer to the table in the "Response to Reviewers" document summarizing author response to reviewer comments. We also attached the revised manuscript in track change and clean version formats for your perusal.

---

## [Decision Letter · Decision Letter 1]

20 Oct 2022

Cost-benefit analysis of a trifocal intraocular lens versus a monofocal intraocular lens from the patient’s perspective in the United States

PONE-D-22-16771R1

Dear Dr. John Palmer Berdahl,

We’re pleased to inform you that your manuscript has been judged scientifically suitable for publication and will be formally accepted for publication once it meets all outstanding technical requirements.

Kind regards,

Guangming Jin, M.D.

Academic Editor

PLOS ONE

Additional Editor Comments (optional):

Reviewers' comments:

Reviewer's Responses to Questions

**Comments to the Author**

1. If the authors have adequately addressed your comments raised in a previous round of review and you feel that this manuscript is now acceptable for publication, you may indicate that here to bypass the “Comments to the Author” section, enter your conflict of interest statement in the “Confidential to Editor” section, and submit your "Accept" recommendation.

Reviewer #1: All comments have been addressed

Reviewer #2: All comments have been addressed

2. Is the manuscript technically sound, and do the data support the conclusions?

Reviewer #1: Yes

Reviewer #2: Yes

3. Has the statistical analysis been performed appropriately and rigorously? 

Reviewer #1: Yes

Reviewer #2: Yes

4. Have the authors made all data underlying the findings in their manuscript fully available?

Reviewer #1: Yes

Reviewer #2: Yes

5. Is the manuscript presented in an intelligible fashion and written in standard English?

Reviewer #1: Yes

Reviewer #2: Yes

6. Review Comments to the Author

Reviewer #1: Not applicable. The authors have satisfactorily addressed all of my comments on the initial draft manuscript.

Reviewer #2: Thank you for the revisions to this article. I have no further recommendations for this manuscript.

7. PLOS authors have the option to publish the peer review history of their article (what does this mean?). If published, this will include your full peer review and any attached files.

Reviewer #1: No

Reviewer #2: No

---

## [Editor Report · Acceptance letter]

25 Oct 2022

PONE-D-22-16771R1 

Cost-benefit analysis of a trifocal intraocular lens versus a monofocal intraocular lens from the patient’s perspective in the United States 

Dear Dr. Berdahl:

I'm pleased to inform you that your manuscript has been deemed suitable for publication in PLOS ONE. Congratulations! Your manuscript is now with our production department. 

Kind regards, 

on behalf of

Dr. Guangming Jin 

Academic Editor

PLOS ONE